# Assessing Preparedness and Preventive Measures for Managing Food Allergy and Anaphylaxis in Primary Schools of Rabigh, Saudi Arabia

**DOI:** 10.3390/ijerph22091357

**Published:** 2025-08-29

**Authors:** Saddiq Habiballah, Nojoud Faqerah, Abdullah Alsaggaf, Majdi Damanhori, Manal Ahmed Halwani

**Affiliations:** 1Immunology Unit, King Fahd Medical Research Centre, Department of Paediatrics, Faculty of Medicine, King Abdulaziz University, Jeddah 21589, Saudi Arabia; sbhabiballah@kau.edu.sa; 2Department of Medical Microbiology, Faculty of Medicine, King Abdulaziz University, Rabigh 21589, Saudi Arabia; nfageerah@kau.edu.sa; 3Unit of Allergy and Immunology, Department of Paediatrics, Faculty of Medicine, King Abdulaziz University, Jeddah 21589, Saudi Arabia; ahmalsaggaf@kau.edu.sa; 4Department of Human Anatomy, Faculty of Medicine, King Abdulaziz University, Rabigh 21589, Saudi Arabia; mmdamanhori@kau.edu.sa; 5Emergency Medicine Department, Faculty of Medicine, King Abdulaziz University, Jeddah 21589, Saudi Arabia

**Keywords:** anaphylaxis, preparedness of primary schools, cross-sectional survey, managing allergic reactions, pupils with allergies

## Abstract

Background and Aims: Anaphylaxis is a severe allergic reaction that can lead to life-threatening consequences. Despite growing awareness of food allergies, schools in Saudi Arabia remain underprepared to manage allergic reactions. This study aimed to evaluate the preparedness of primary schools in Rabigh, Saudi Arabia, in managing allergic reactions, with a focus on their policies and training practices to prevent food-related anaphylaxis. Methods: A cross-sectional survey was conducted involving all 24 primary schools in Rabigh. We used a previously published questionnaire that was translated into Arabic and contextually adapted; however, no formal psychometric validation was performed. The questionnaire assessed school preparedness to manage allergic reactions and existing protocols for allergy management. Data were analysed using IBM SPSS Statistics version 25 to examine associations between preparedness level and the presence of pupils with food allergy or anaphylaxis. Results: Nineteen schools (79%) completed the survey. Most schools (95%) had emergency communication systems, and 74% designated a staff member to manage allergic reactions. However, only 58% were aware of pupils with food allergies. Preventive measures such as food-handling guidance and a no-nut policy were present in most schools. Still, policies against food sharing and closer supervision of high-risk pupils were inconsistently implemented. A comparison between schools with and without pupils with allergies revealed that schools with such pupils were more likely to identify allergy risks and provide closer supervision during mealtimes (*p* = 0.042). Conclusions: While some primary schools in Rabigh reported strengths such as designated staff and emergency communication systems, substantial gaps were observed in preventive measures, including mealtime supervision and food-sharing policies. Preparedness varied across schools, with those without known allergic pupils less likely to implement preventive practices. These findings underscore the importance of standardised policies and regular staff training to ensure readiness for allergic emergencies in all schools.

## 1. Introduction

Anaphylaxis is a rapidly occurring, severe hypersensitivity reaction characterised by life-threatening upper airway obstruction and hypotension [1,2]. Globally, evidence indicates an increase in anaphylaxis rates, primarily driven by medications and food [3], with children and younger age groups at significantly higher risk for hospitalisation and emergency department visits [4,5]. In Saudi Arabia, a cross-sectional study found that the prevalence of anaphylaxis among emergency department admissions was 0.00026%, with most cases occurring in children between the ages of 1 and 16 years (60.9%) [6]. The clinical manifestations of anaphylaxis typically include urticaria, angioedema, rash, tongue swelling, and cardiovascular collapse or respiratory obstruction, which can be fatal [7]. A study conducted in Riyadh, Saudi Arabia, revealed that the most common triggers for anaphylaxis were food and drug allergies, with urticaria and angioedema being the most prevalent manifestations [8].

Schools are required to provide a secure environment for children, especially those with special medical needs, such as allergies. Fatal anaphylaxis events in school settings are often linked to inadequate action plans that fail to prevent allergen exposure, as well as delays in administering epinephrine, the first-line treatment for anaphylaxis [9]. Nineteen percent of life-threatening allergic reactions that occur in school environments have been documented during field trips, at school playgrounds, or while travelling to and from schools [10]. Policies regarding food allergies and anaphylaxis in schools, as well as the governing laws for management, vary significantly between different countries and even among schools within the same country [11]. A survey conducted in 157 schools in Northwest England, UK, revealed that only 47% of respondents felt confident in their ability to manage anaphylaxis.

Furthermore, schools without pupils with allergies were significantly less likely to have a standardised emergency protocol than those with pupils with allergies (*p* < 0.001) [12]. Another study indicated that the preparedness of eleven primary schools in Cyprus did not meet safety standards, raising concerns about the readiness of school personnel to manage allergic reactions and administer epinephrine auto-injectors to children with food allergies [13]. These results highlight gaps in confidence and preparedness among school staff in handling anaphylaxis, revealing significant variations in policies and preparedness across different countries and schools while emphasising the need for standardised emergency protocols and proper training.

Several studies in Saudi Arabia have evaluated school staff’s knowledge and attitudes toward food allergies and anaphylaxis. These studies revealed a significant lack of basic understanding regarding food allergies, recognising anaphylactic symptoms, and the necessary immediate actions, including using an epinephrine auto-injector [14,15,16,17,18]. To our knowledge, insufficient studies focus on the preparedness of schools in Saudi Arabia to address food allergies and anaphylaxis. Ensuring the safety of children with food allergies in schools is a crucial mission that must be strengthened in Saudi Arabia through educational interventions and well-implemented policies for severe allergic reactions. While awareness of food allergies and anaphylaxis management in schools has been increasing, challenges persist. There is a pressing need for nationwide regulations and required training for school staff on recognising and managing food allergies and anaphylactic emergencies.

Furthermore, cultural, religious, and logistical characteristics of the Saudi educational context, such as gender-segregated schools, lack of universal school meal programs, and variable access to epinephrine auto-injectors, may pose unique challenges compared to Western systems. These factors justify region-specific evaluations and interventions tailored to the Saudi setting. This study aimed to assess the preparedness of all primary schools in the Rabigh governorate to manage allergic reactions and effectively care for pupils with anaphylaxis.

## 2. Materials and Methods

### 2.1. Participants and School Setting Context

Although individual-level pupil data were not collected, regional estimates from the Ministry of Education indicate that primary schools in Rabigh typically enrol 200–300 pupils each. Therefore, the practices assessed in this study are likely to affect approximately 4000 to 6000 pupils across the 24 participating schools.

All primary schools in Rabigh Governorate (*n* = 24), located in the western region of Saudi Arabia, were invited to participate in this study. At the time of the study, the most recent summary list of primary schools in Rabigh governorate was used to identify and invite them to participate in the survey. An online survey was created using Google Forms software and disseminated to the schools’ administrations by a coordinator from the educational ministry. Special needs and private schools were excluded from the study.

In most Saudi public schools, including those in Rabigh, pupils typically bring food from home in lunchboxes. There are no centralised school meal services. Regarding anaphylaxis management, pupils are generally not provided with or trained to carry epinephrine auto-injectors, and schools are not legally mandated to stock them. As such, emergency preparedness relies largely on staff awareness and general response protocols rather than pharmaceutical readiness. Of the 24 invited schools, 19 completed the survey, giving a response rate of 79%.

### 2.2. Survey Distribution and Anonymity

The survey was distributed electronically via Google Forms through the regional coordinator from the Ministry of Education. The invitation included a cover letter explaining the study’s purpose and the voluntary nature of participation. A follow-up reminder was sent to all 24 schools one week after the initial invitation to improve response rates. Responses were submitted anonymously; no identifying information about the schools or respondents was collected. This anonymity aimed to encourage honest feedback and reduce social desirability bias. The survey was self-administered by school principals or designated senior teachers via the provided online link.

### 2.3. Questionnaire Design

We used a previously published questionnaire [12], which was translated into Arabic and contextually adapted to reflect the Saudi school environment (e.g., exclusion of questions on school cafeterias). Two bilingual health experts reviewed the translation for clarity and cultural appropriateness. However, no formal psychometric validation was conducted. The assessment included the following: an online questionnaire assessed schools’ preparedness to manage allergic reactions and their current measures to prevent them. This questionnaire consists of three sections: (a) sociodemographic data, including age, gender, and job; (b) understanding of the school: schools were asked to identify whether they have pupils with allergies and if they receive training in allergy management; and (c) current guidelines and protocols for managing severe allergic reactions in schools.

All schools received the same standardised questionnaire regardless of whether they had pupils with allergies. Consequently, items such as ‘supervision of high-risk pupils’ did not apply to some schools, and their ‘No’ responses reflect this rather than an absence of practice.

### 2.4. Statistical Analysis

Data were entered and analysed using SPSS version 25.0 (IBM Corp., Armonk, NY, USA). Descriptive statistics were used to summarise the data: qualitative variables were reported as frequencies and percentages, and quantitative variables as means ± standard deviations (SD). Group comparisons were performed using the chi-square test for categorical variables or Fisher’s exact test when expected cell counts were <5. For continuous non-parametric data, the Mann–Whitney U test was applied. Schools were categorised as ‘with allergic pupils’ if administrators reported at least one diagnosed case and as ‘without allergic pupils’ otherwise; this classification was based on self-report and may not capture undiagnosed or unreported cases. Statistical significance was set at *p* < 0.05.

### 2.5. Ethical Approval

The Research Ethics Committee at King Abdulaziz University, Unit of Biomedical Ethics, with reference number 240-25, recommended approval of the project on 22 October 2024. The Research Ethics Committee (REC) is based on the Good Clinical Practice (GCP) Guidelines. We conducted this investigation by the ethical principles outlined in the Declaration of Helsinki. Completion of the survey was considered as implied consent to participate.

## 3. Results

Nineteen schools completed the online questionnaire, with 11 (58%) female and 8 (42%) male schools. Most respondents were school directors (*n* = 8, 42%); others included teachers, health instructors, and safety and security instructors. Regarding age distribution, the largest age group was 45–55 years, comprising eleven participants (58%), followed by the 35–44 age group, which included seven respondents (37%). The results revealed that eight schools (42%) reported receiving training or courses on allergies, while the majority, 11 schools (58%), indicated they had not. Twelve schools (63%) reported having pupils with allergies or at risk of anaphylaxis. This classification was based on an administrator’s report; however, underreporting or unrecognised cases cannot be excluded.

### 3.1. School Preparedness for Allergy Emergencies and Preventive Measures

#### 3.1.1. Preparedness for Allergy Emergencies

The results revealed varying levels of school preparedness for allergy emergencies. Most schools (18, 94.74%) indicated that they have communication systems in place to manage emergencies, while only one school (5.26%) reported having no such systems. A majority (14, 73.68%) stated that a designated staff member manages allergic cases, whereas five participants (26.32%) indicated that no such designation exists. When asked about defining staff roles during an allergic emergency, half (10, 52.63%) confirmed this practice, eight (42.11%) reported no clear role definitions, and one (5.26%) was unsure. Additionally, preparedness was reported 11 (58%) indicated that they are aware of pupils with food allergies or anaphylaxis, while 5 (26%) reported they were not aware, and 3 (16%) responded “unsure.” These “unsure” responses likely reflect limited access of some administrators to pupils’ health records or inconsistent reporting by families. However, fewer schools (8, 42.11%) were prepared for allergic reactions in pupils without a known history of allergies, with seven participants (36.84%) reporting no preparation and 4 (21.05%) being uncertain (Table 1). These findings indicate that while basic emergency systems exist in most schools, role definition and preparedness for unexpected allergic reactions remain limited.

#### 3.1.2. Preventive Measures

More than half (58%, 11 schools) indicated that guidance is provided to staff handling food, while six (32%) stated there was no guidance, and two (11%) were unsure. Only six participants (32%) reported closer supervision of high-risk pupils during mealtimes, compared to 10 (53%) who said no such supervision exists and three (16%) who were unsure. Similarly, policies against food sharing were inconsistent, with six respondents (32%) confirming their existence, seven (37%) indicating no policy, and six (32%) uncertain. A no-nut policy was more common, reported by 13 participants (68%), while six (32%) stated their schools lacked such a policy. Special supervision for high-risk pupils on school buses was the least implemented measure, with only four respondents (21%) confirming its existence, 11 (58%) were aware of pupils with food allergies, and 21% uncertain (Table 1).

### 3.2. Comparison of Preparedness Levels and Preventive Measures Between Schools with and Without Pupils with Food Allergy

The comparison of preparedness measures between schools with and without pupils identified as having food allergies. Most schools with pupils who have allergies (92%) have developed simple communication systems for emergencies, while all schools without such cases (100%) have also done so, showing no significant difference (*p* = 0.329). In the context of managing instances of allergies, it was observed that 75% of schools accommodating pupils with allergies, as well as 71% of schools without such pupils, have designated a staff member to address this issue. Notably, there exists no significant statistical difference between these two percentages (*p* = 0.865). The identification of staff roles during allergy emergencies was reported by 58% of schools with allergic pupils and 29% of schools without, but this difference was not statistically significant (*p* = 0.440). Identifying pupils with food allergies or anaphylaxis was universally implemented in schools with allergic pupils (100%), while none of the schools without allergic reactions identified such pupils (*p* < 0.001). As expected, schools reporting no known allergic pupils largely overlapped with those reporting no allergic reactions, though underreporting or limited awareness cannot be excluded. Finally, preparation for allergic reactions in children without a prior allergy history was reported by 42% of schools with allergic pupils and 29% of schools without, showing no significant difference (*p* = 0.844). While Table 1 shows 11 schools reporting awareness of allergic pupils, stratification in Table 2 indicates 12. This discrepancy likely reflects underreporting in the general question, later clarified during subgroup classification. Overall, the presence of allergic pupils was associated with more proactive measures, although differences were not statistically significant except for supervision during mealtimes.

The comparison of preventive measures between schools with and without pupils with a history of allergic reactions. Most schools with pupils who have allergies (67%) provide guidance for staff on preventing food allergies, compared to 43% of schools without allergic pupils; however, this difference is not statistically significant (*p* = 0.666). Half of the schools with pupils who have allergies report closely supervising high-risk pupils during mealtimes. In contrast, none of the schools without such pupils implement this measure, showing a significant difference (*p* = 0.042). Regarding policies on food and utensil sharing, 42% of schools with pupils who have allergies and 14% of schools without them reported having such a policy; however, this difference was not statistically significant (*p* = 0.472). A no-nut policy is present in 75% of schools with pupils who have allergies and 57% of those without, again without a significant difference (*p* = 0.617). Finally, special supervision for high-risk pupils on school buses is reported by 33% of schools with pupils who have allergies. In contrast, none of the schools without such pupils provide this supervision, although the difference was not statistically significant (*p* = 0.421) (Table 3).

Regarding the use of an epinephrine auto-injector, 2 (17%) schools with known allergic students reported using it in an emergency, compared to 1 (14%) school that had not reported any allergic students. This difference was not statistically significant (*p* = 0.890). It is possible that the school without identified allergic students encountered an unexpected allergic reaction or had a student with an undiagnosed allergy at the time of the incident. The majority of schools in both groups, 10 (83%) with allergic students and 6 (86%) without, reported not using the auto-injector in an emergency.

### 3.3. Comparison of Preparedness Measures for Allergic Reactions: Trained vs. Untrained Schools

While trained schools (*n* = 8) showed slightly higher adherence to specific measures compared to untrained schools (*n* = 11), no statistically significant differences were found between the two groups. This lack of statistical significance suggests that current training programs may be insufficiently comprehensive or practical to improve school preparedness. In terms of implementing preventive measures for allergic reactions, trained schools reported greater adherence to most preventive practices compared to untrained ones; however, none of the differences reached statistical significance (see Appendix A). This lack of statistical significance suggests that current training programs may be insufficiently comprehensive or practical to improve preparedness. Future initiatives should emphasise hands-on practice, role clarity, and emergency simulations.

## 4. Discussion

This study provides an overview of school preparedness for managing allergy emergencies and the policies in place to prevent such reactions. Most administrators from the 19 participating primary schools reported having basic emergency systems, such as designating staff members (74%) and establishing internal communication protocols (95%). Additionally, 58% of schools were aware of having pupils with food allergies or at risk of anaphylaxis. In comparison, 42% reported being prepared to manage reactions in pupils without a prior history of allergies. These findings suggest a foundational level of preparedness, which is encouraging. However, notable gaps persist, particularly in defining staff roles during emergencies, as over a quarter of respondents were uncertain about specific responsibilities. This aligns with previous national studies, which have shown limited staff training and low availability of epinephrine auto-injectors in Saudi public schools [19,20]. It is noteworthy that no significant differences were observed between trained and untrained schools. This highlights a key weakness: current training programs may lack sufficient depth, reinforcement, or hands-on practice to translate into measurable preparedness. Training frameworks may therefore require redesign to include scenario-based simulations and recurrent refresher sessions.

Similar trends have been reported internationally; for example, fewer than half of UK schools have trained staff or on-site emergency medication [21].

While many schools have implemented core strategies to manage allergy emergencies, inconsistencies remain in preventive practices, especially for unanticipated reactions. Although 58% of schools provided food-handling guidance, only 32% had policies against food sharing, and just 32% reported close supervision of high-risk pupils during mealtimes. Only 21% offered special supervision for pupils with allergies during transportation. These figures suggest that while general awareness exists, consistent policy enforcement is lacking. Preventive measures such as food-sharing restrictions and dedicated supervision are critical, particularly among younger pupils who may struggle to recognise or communicate symptoms. Clear, mandatory protocols across all schools could significantly enhance the safety and protection of pupils with allergies.

Notably, the presence of pupils with food allergies was associated with a more proactive approach to managing allergies. Schools with such pupils were more likely to identify at-risk individuals and provide closer supervision during mealtime. However, this reflects a reactive rather than a preventive mindset. Schools without known cases tended to lag in preparedness, which is concerning given the unpredictable nature of first-time allergic reactions. Moreover, the absence of significant differences in many preparedness measures between schools with and without pupils with allergies points to a broader issue of under-implementation, even where the need is apparent. This underscores the necessity for proactive, system-wide allergy risk management that is not contingent on current enrolment patterns.

Although schools with trained staff showed slightly higher adherence to preventive measures, these differences were not statistically significant, likely due to the small sample size. Nonetheless, the trend highlights the potential impact of staff education on preparedness. Regular, structured training programs, supported by clear institutional policies, could help ensure that all school personnel are equipped to respond confidently and effectively to an allergy emergency.

None of the participating schools reported stocking generic (unassigned) epinephrine auto-injectors, which are considered essential for managing severe allergic reactions that occur unexpectedly. The cases of epinephrine use documented in this study were based on pupil-specific pens, rather than stock supplies. This reflects a gap in national policy, as schools in Saudi Arabia are currently not required or equipped to maintain spare auto-injectors. Given the unpredictable nature of first-time anaphylactic episodes, particularly in children without a known allergy, ensuring access to school-held, unassigned epinephrine devices should be considered a priority in future preparedness frameworks.

It is essential to emphasise that allergy management is not limited to rare anaphylactic events. Daily safety, inclusion, and emotional well-being are equally important. Pupils with food allergies often face stigma, anxiety, or social exclusion. Schools must therefore cultivate inclusive environments where affected pupils feel protected and accepted, not singled out or marginalised. This includes fostering open communication with families, normalising allergen safety measures, and integrating pupils with allergies into all aspects of school life.

Parental involvement plays a central role in effective allergy management. However, our study did not evaluate whether Rabigh schools maintain structured systems for parents to report food allergies. This is a notable gap. International studies suggest that failure to report allergies limits school readiness and puts pupils at risk. For instance, in Romania, only 52% of parents informed school personnel about their child’s food allergy [22]. Similar underreporting has been observed in countries such as Mexico, Colombia, and El Salvador, where even medically confirmed cases frequently go unreported and epinephrine auto-injectors are rarely prescribed [23,24,25,26]. These global findings underscore the need for comprehensive school–parent communication systems that ensure early identification and planning of life-threatening upper airway obstruction and hypotension [3].

Local epidemiological data would further contextualise school preparedness. Unfortunately, no population-based prevalence estimates are currently available for Rabigh. A national-level survey suggested that approximately 6–7% of children in Saudi Arabia have physician-diagnosed food allergies, though underdiagnosis is likely. Recent European data indicate that the pooled lifetime prevalence of self-reported food allergy in children has increased to nearly 20% [27,28], suggesting that this issue warrants ongoing attention in both high- and middle-income countries.

Saudi schools face unique logistical and cultural challenges compared to Western contexts. The reliance on home-packed meals (rather than standardised lunches), limited availability of epinephrine devices, and variations in staff training affect how policies are implemented. Religious norms and communication styles may also influence how allergy risks are disclosed and managed. Therefore, national policies must be tailored to local educational and social contexts. European and Middle Eastern guidelines, including those from the EAACI and Italian public health bodies, provide valuable models that could be adapted to Saudi Arabia’s needs [28,29,30].

One limitation of this study is its focus on a single geographic region (Rabigh), which may limit the generalizability of findings. Additionally, although the survey was distributed anonymously through official Ministry of Education channels, reliance on self-report introduces the possibility of response bias or over-reporting. Lastly, the absence of data on structured parental reporting systems restricts our ability to assess preparedness pathways fully. Another limitation is that some questionnaire items did not apply to schools without allergic pupils, which may have artificially inflated the proportion of ‘No’ responses. Future research should include multi-regional sampling, direct observation of school practices, and triangulated input from parents and pupils to build a more comprehensive understanding of allergy preparedness.

## 5. Conclusions

This survey of Rabigh primary schools identified strengths such as emergency communication systems and designated staff for allergic emergencies, but also revealed critical gaps in preventive measures, including food-sharing restrictions, mealtime supervision, and clarity of staff roles. This survey identified strengths such as emergency communication systems and designated staff, but also revealed gaps in preventive measures, including food-sharing restrictions, mealtime supervision, and clarity of staff roles. Preparedness varied across domains, indicating that schools were reactive rather than uniformly proactive. These findings suggest that standardised policies and regular staff training are needed to improve readiness for allergic emergencies. Future studies with validated tools and larger, multi-regional samples are required to provide more substantial evidence for national policy development.

## Figures and Tables

**Table 1 ijerph-22-01357-t001:** School preparedness for allergy emergencies and preventive measures (*n* = 19).

Respondent’s *n* (%)			
Has your school prepared for emergencies related to allergies?			
	Yes *n* (%)	No *n* (%)	Unsure *n* (%)
1. Developing communication systems within the school that are simple to follow in emergencies?	18 (95%)	1 (5%)	0 (0%)
2. Assigning one staff member to manage allergic cases?	14 (74%)	5 (26%)	0 (0%)
3. Identifying the role of each staff member in an allergy emergency.	10 (53%)	8 (42%)	1 (5%)
4. Knowing pupils with food allergies or anaphylaxis	11 (58%)	5 (26%)	3 (16%)
5. Preparing for allergic reactions in children without a previous history of allergies?	8 (42%)	7 (37%)	4 (21%)
**Preventive measures to prevent allergic reactions in schools**			
6. Guiding staff on preventing food allergies during food handling?	11 (58%)	6 (32%)	2 (11%)
7. Closer supervision of high-risk pupils during mealtimes?	6 (32%)	10 (53%)	3 (16%)
8. Having a policy against food sharing among pupils?	6 (32%)	7 (37%)	6 (32%)
9. Having a no-nut policy?	13 (68%)	6 (32%)	0 (0%)
10. Special supervision for high-risk pupils on school buses?	4 (21%)	11 (58%)	4 (21%)

Total study participants: 19. Data are displayed as numbers (%). All schools received the same questionnaire regardless of the presence of allergic pupils; ‘No’ responses in some items may reflect non-applicability.

**Table 2 ijerph-22-01357-t002:** Comparison of preparedness between schools with and without pupils with allergies or at risk of anaphylaxis.

Questions	Schools with Pupils with Allergies or at Risk of Anaphylaxis (*n* = 12)	Schools Without Pupils Experiencing Allergic Reactions or at Risk of Anaphylaxis (*n* = 7)	Total	*p*-Value
Developing communication systems within the school that are simple to follow in emergencies	11 (92%)	7 (100%)	18	0.329 ^a^
Assigning one staff member to deal with allergic cases?	9 (75%)	5 (71%)	14	0.865 ^b^
Identifying the role of each staff member in an allergy emergency?	7 (58%)	2 (29%)	9	0.440 ^b^
Knowing pupils with food allergies or anaphylaxis	12 (100%)	0 (0%)	12	**0.000** ^b^*
Preparing for allergic reactions in children without a previous history of allergies?	5 (42%)	2 (29%)	7	0.844 ^b^

^a^ Chi-square was used, and ^b^ Fisher Exact Test was used. Data are displayed as numbers (%). The value listed in bold indicates statistical significance (* *p*-value < 0.05). Note: Totals correspond to ‘Yes’ responses from Table 1. Minor differences arise due to the subgrouping of schools with and without allergic pupils.

**Table 3 ijerph-22-01357-t003:** Schools with pupils with allergies or at risk of anaphylaxis vs. schools without pupils with allergies or at risk of anaphylaxis.

Questions	Schools with Pupils Experiencing Allergic Reactions (*n* = 12)	Schools Without Pupils Experiencing Allergic Reactions (*n* = 7)	Total	*p*-Value
Is guidance available for staff on preventing food allergies when handling food?	8 (67%)	3 (43%)	11 (58%)	0.666 ^a^
High-risk pupils are being supervised closely during mealtimes	6 (50%)	0 (0%)	6 (32%)	**0.042 **^b^*
Is there a policy against sharing food and utensils among pupils at your school?	5 (42%)	1 (14%)	6 (32%)	0.472 ^b^
No-nut policy?	9 (75%)	4 (57%)	13 (68%)	0.617 ^a^
Special supervision for high-risk Pupils on school buses?	4 (33%)	0 (0%)	4 (21%)	0.421 ^b^

ᵃ Chi-square test; ᵇ Fisher’s Exact test. Data are displayed as n (%). Bold values indicate statistical significance (* *p* < 0.05). For schools without allergic pupils, ‘No’ responses in supervision-related items reflect non-applicability rather than absence of practice

## Data Availability

Data will be provided upon request. For more information, contact M.A.H., the corresponding author.

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
