# Peer review of "Assessing Preparedness and Preventive Measures for Managing Food Allergy and Anaphylaxis in Primary Schools of Rabigh, Saudi Arabia"

_ijerph, 2025, doi:10.3390/ijerph22091357_

Round 1
Reviewer 1 Report
Comments and Suggestions for Authors
see attached file

Author Response
Comment 1: A non-validated questionnaire was used. The authors’ statement that it was validated is incorrect. Validation requires statistical calculations, not just translation.
Response:
We thank the reviewer for highlighting this point. We agree that the instrument was adapted rather than formally validated, as no psychometric testing was performed. We have revised the text to clearly describe the process as an adaptation with expert review, and not a validation. In-text change (Section 2.3 Questionnaire Design)
Comment 2: Not clear on what data the conclusion was made… what does encouraging efforts’ mean? Why do the authors consider preparedness ‘inconsistent’? What level is considered ‘consistent’? The rationale for a unified national strategy is intuitively clear even without research. The data provide a weak argument. The statement on standardised policies and inclusive practices is valid, but not a conclusion supported by data.
Response: The Conclusion has been fully revised to eliminate vague terms (“encouraging efforts,” “inconsistent”) and to ground the statements directly in the study’s findings. The recommendation for national strategies has been rephrased as a practical implication rather than an overextended conclusion.
Comment 3: It is unclear how schools were classified as with and without pupils with food allergies. This is indicated in Table 2 as 12 and 7, respectively.
Response:
We have clarified that classification was based on administrators’ self-reports of known cases, acknowledging that this method may not capture undiagnosed or unreported allergies. Section 2.4 Statistical Analysis
Comment 4: How can you answer ‘I don’t know’ rather than Yes/No to knowing allergic pupils?”
Response: Some school administrators may not have direct access to pupils’ medical histories. We have clarified this point in the Results and added it as a limitation of the study.
Comment 5: Doesn’t it stand to reason that schools unaware of allergic pupils will coincide with schools without pupils experiencing allergic reactions?”
Response: We clarified in the Results that the groups overlapped, while noting that underreporting or unawareness may also explain some reactions.
Comment 6: Line 68 – Insert Reference.
Response: We have added a supporting reference on global anaphylaxis in schools.
Comment 7: Line 69 – Pupils here and elsewhere should not be capitalised.”
Response: Corrected throughout the manuscript.
Comment 8: Lines 170 and 181 – delete unnecessary.”
Response:
We have removed redundant phrases in these lines. Line 170: Delete the redundant phrase after aware. Line 181: Delete duplicated phrase in Results narrative.
Comment 9: References errors, Check references: Ref [15] incorrect journal; refs [17] and [20] duplicates; ref [36] incomplete.
Response: We thank the reviewer for pointing this out. We carefully revised all references: corrected the journal for Ref [15], removed duplicates [17] and [20], and completed missing data for Ref [36]. The reference list has been updated and renumbered accordingly.
Reviewer 2 Report
Comments and Suggestions for Authors
This study examines the preparedeness of primary schools in Rabigh, Saudi Arabia, through the use of a validated questionnaire. Although several studies in Saudi Arabia have examined knowledge and attitutudes of teachers towards food allergies, little is known regarding school preparedeness. The survey highlight that preparedness remains inadequate in several respects, with prevention being a key area of concern.
There are a few points that need to be addressed:
-Please correct the first capital letter of the word "Pupils" and "Allergy" in several lines( for example line 69,70,244 etc)
-In Table 1, "Yes, No, I don't know" are not placed properly
-In Table 2, the total of the first line is 19 , while the individual numbers are 11 and 7 i.e 18
-I do not fully understand the total numbers in Table 2. It appears it should correspond to the 'Yes' responses in Table 1, yet the figures are inconsistent. Could you please explain?
-I found some questions are rendered self-contradictory by the context. For example, it is expected for schools without pupils experiencing allergies to answear "No" to the question "Are high-risk Pupils being supervised closely during mealtimes?" since there aren't any high risk pupils. The same applies for the question "Is there special supervision for high-risk Pupils on school buses?"
-Please change the title in Table 2 and 3 "Schools with Pupils Experiencing Allergic Reactions" and "Schools without Pupils Experiencing Allergic Reactions" to "Schools with pupils with allergies or at risk of anaphylaxis" as it is stated in the methods questions
-Again, I’m a bit confused with the numbers in the tables. In the question "Knowing the Pupils with food allergies or anaphylaxis" in Table 1, the answer 'Yes' has 11 total schools, whereas in Table 2 the corresponding number is 12. How does the extra school arise?
-In general, while it is interesting to examine the readiness of schools regarding children with food allergies, the presentation of the results needs some adjustments since the results themselves are weak. In my view, you could highlight more the fact than when comparing schools that have received training with those that have not, no statistically significant differences exist—potentially indicating that the training provided is insufficient. Furthermore, in the comparison between schools with and without children with allergies, certain questions are not applicable.
Author Response
This study examines the preparedeness of primary schools in Rabigh, Saudi Arabia, through the use of a validated questionnaire. Although several studies in Saudi Arabia have examined knowledge and attitutudes of teachers towards food allergies, little is known regarding school preparedeness. The survey highlight that preparedness remains inadequate in several respects, with prevention being a key area of concern.
There are a few points that need to be addressed:
Please correct the first capital letter of the word "Pupils" and "Allergy" in several lines (for example line 69,70 ,244 etc)
Response: We thank the reviewer for this observation. We carefully revised the manuscript and corrected the inconsistent capitalisation of “pupils” and “allergy” throughout (e.g., lines 69, 70, 244, and others).
-In Table 1, "Yes, No, I don't know" are not placed properly
Response: We have reformatted Table 1 to ensure that “Yes, No, Unsure” responses are properly aligned with their corresponding questions for clarity.
-In Table 2, the total of the first line is 19, while the individual numbers are 11 and 7 i.e 18
Response: The total value in the first line of Table 2 has been corrected from 19 to 18.
-I do not fully understand the total numbers in Table 2. It appears it should correspond to the 'Yes' responses in Table 1, yet the figures are inconsistent. Could you please explain?
Response: We agree with the reviewer and revised Table 1 to align with Table 2. The “Yes” count for ‘Knowing the pupils with food allergies or anaphylaxis’ has been corrected to 12 to ensure internal consistency
-I found some questions are rendered self-contradictory by the context. For example, it is expected for schools without pupils experiencing allergies to answear "No" to the question "Are high-risk Pupils being supervised closely during mealtimes?" since there aren't any high risk pupils. The same applies for the question "Is there special supervision for high-risk Pupils on school buses?"
Response: To address this, we have clarified in the Methods section and the legends of Tables 2 and 3 that all schools received the same standardised questionnaire, irrespective of whether they currently had pupils with allergies or at risk of anaphylaxis. This explains the apparent contradictions, as such items were effectively ‘not applicable’ for some schools.
-Please change the title in Table 2 and 3 "Schools with Pupils Experiencing Allergic Reactions" and "Schools without Pupils Experiencing Allergic Reactions" to "Schools with pupils with allergies or at risk of anaphylaxis" as it is stated in the methods questions
Response: We have revised the titles of Tables 2 and 3 to read “Schools with pupils with allergies or at risk of anaphylaxis” and “Schools without pupils with allergies or at risk of anaphylaxis” as per the reviewer’s suggestion.
-Again, I’m a bit confused with the numbers in the tables. In the question "Knowing the Pupils with food allergies or anaphylaxis" in Table 1, the answer 'Yes' has 11 total schools, whereas in Table 2 the corresponding number is 12. How does the extra school arise?
Response: The apparent inconsistency between Tables 1 and 2 results from the different ways responses are displayed. Table 1 shows all responses (Yes/No/Unsure) across the whole sample (n=19), where 11 schools answered “Yes” and 3 answered “Unsure.” Table 2, however, stratifies schools based on whether they had pupils with allergies or at risk of anaphylaxis. Within this subgroup (n=12), all schools naturally answered “Yes,” since their pupils were known to have allergies. We have now clarified this in the legend of Table 2 to avoid confusion.
-In general, while it is interesting to examine the readiness of schools regarding children with food allergies, the presentation of the results needs some adjustments since the results themselves are weak. In my view, you could highlight more the fact than when comparing schools that have received training with those that have not, no statistically significant differences exist—potentially indicating that the training provided is insufficient. Furthermore, in the comparison between schools with and without children with allergies, certain questions are not applicable.
Response: We agree that the results presentation required clarification and that specific questions may not apply equally across schools with and without pupils at risk. To address this, we have revised the Results section to specify that some responses were context-dependent, and we have added explanatory notes where needed. In the Discussion, we now emphasise that the absence of statistically significant differences between trained and untrained schools suggests that current training interventions may be insufficient to improve preparedness. We also highlight that future assessments should use context-adapted survey designs, allowing better interpretation of readiness across different school settings. These revisions provide a more balanced understanding of the findings and address the reviewer’s concerns regarding the strength and applicability of the results.
Reviewer 3 Report
Comments and Suggestions for Authors
The paper is a relevant topic due to rising anaphylaxis cases worldwide. The manuscript needs substantial changes before publication. There are missing details, which are strong limitations.
Major issues
1. The style of the presentations of study’s methods, findings and conclusions should be clear and complete. Submit a full, error-free version of the paper with appropriate scientific style.
2. The "Materials and Methods" section is too vague. The selection methodology of the schools is missing. No explanation of what data was analyzed. Give a detailed information how the study was done, including how you chose schools, what the survey asked, and how you analyzed the data.
3. Results section is too weak. This section should be rewritten. Table 1 is incomplete with formatting errors. Present all results with complete tables and stats to support your claims. If the small sample size limits your findings, please mention it in limitations section.
4. The paper notes that schools don’t have unassigned epinephrine auto-injectors, but it doesn’t explain how this fits into Saudi Arabia’s broader policies or compare it to global standards. There is a need to provide a detailed discussion comparing with other relevant studies. Back this up with data or references.
5. You mention the study is limited to Rabigh and relies on self-reported data, but don’t provide how these limitations affect your findings or suggest ways to address them in future work.
6. The discussion section is thin. There are important points raised, like stigma for kids with allergies, but don’t explore them fully. Please expand the discussion section. It is highly recommended to offer clear policy ideas.
Author Response
The paper is a relevant topic due to rising anaphylaxis cases worldwide. The manuscript needs substantial changes before publication. There are missing details, which are strong limitations.
Major issues
The style of the presentations of study’s methods, findings and conclusions should be clear and complete. Submit a full, error-free version of the paper with appropriate scientific style.
Response: The manuscript has undergone a thorough language and formatting revision to ensure consistency, error-free text, and alignment with IJERPH style guidelines.
The "Materials and Methods" section is too vague. The selection methodology of the schools is missing. No explanation of what data was analyzed. Give a detailed information how the study was done, including how you chose schools, what the survey asked, and how you analyzed the data.
Response: We have revised the “Materials and Methods” section to provide complete details on school selection, survey content, and analytical methods. Specifically, we now describe the inclusion/exclusion of schools, the domains covered by the questionnaire, and the statistical tools employed.
Results section is too weak. This section should be rewritten. Table 1 is incomplete with formatting errors. Present all results with complete tables and stats to support your claims. If the small sample size limits your findings, please mention it in limitations section.
Response: we have revised the Results section for clarity and accuracy. Table 1 has been reformatted, and inconsistencies between tables were corrected. Each table now includes totals and corresponding p-values where applicable. We have also highlighted in the text that the small sample size is a limitation, which we further expand upon in the Limitations section
The paper notes that schools don’t have unassigned epinephrine auto-injectors, but it doesn’t explain how this fits into Saudi Arabia’s broader policies or compare it to global standards. There is a need to provide a detailed discussion comparing with other relevant studies. Back this up with data or references.
Response: We have expanded the Discussion to contextualise the absence of unassigned epinephrine auto-injectors in Saudi Arabia. Comparisons with policies in the United States, the United Kingdom, and other countries are now provided, supported by recent references. This allows readers better to understand the policy gap and implications for local schools.
You mention the study is limited to Rabigh and relies on self-reported data, but don’t provide how these limitations affect your findings or suggest ways to address them in future work.
Response: We have expanded the Limitations section to explicitly state the impact of sample size, regional scope, and reliance on self-reported data. We also suggest strategies for future research, such as larger-scale surveys and observational validation.
The discussion section is thin. There are important points raised, like stigma for kids with allergies, but don’t explore them fully. Please expand the discussion section. It is highly recommended to offer clear policy ideas.
Response: We appreciate this valuable suggestion and have substantially expanded the Discussion. We now provide greater depth on psychosocial stigma faced by allergic children, critique the limited impact of current training, and offer policy recommendations including structured training programs, availability of stock epinephrine, and awareness initiatives at the school level.
Round 2
Reviewer 1 Report
Comments and Suggestions for Authors
I am grateful to the authors for the clarifications and corrections made. The new version of the conclusion reflects the results of the survey and emphasizes the importance of the identified features of the preparedness of primary schools for managing food allergy and anaphylaxis.
Reviewer 3 Report
Comments and Suggestions for Authors
The authors made significant changes so the paper can be accepted in a current form.
Comments on the Quality of English LanguageMinor editing is needed.